# Effects of Different Gene Editing Modes of CRISPR/Cas9 on Soybean Fatty Acid Anabolic Metabolism Based on *GmFAD2* Family

**DOI:** 10.3390/ijms24054769

**Published:** 2023-03-01

**Authors:** Junming Zhou, Zeyuan Li, Yue Li, Qiuzhu Zhao, Xinchao Luan, Lixue Wang, Yixuan Liu, Huijing Liu, Jun Zhang, Dan Yao

**Affiliations:** 1College of Life Sciences, Jilin Agricultural University, Changchun 130118, China; 2College of Agronomy, Jilin Agricultural University, Changchun 130118, China

**Keywords:** soybean, fatty acids, CRISPR/Cas9, GmFAD2, editing vector

## Abstract

Δ^12^-fatty acid dehydrogenase (FAD2) is the essential enzyme responsible for catalyzing the formation of linoleic acid from oleic acid. CRISPR/Cas9 gene editing technology has been an essential tool for molecular breeding in soybeans. To evaluate the most suitable type of gene editing in soybean fatty acid synthesis metabolism, this study selected five crucial enzyme genes of the soybean *FAD2* gene family—*GmFAD2-1A*, *GmFAD2-1B*, *GmFAD2-2A*, *GmFAD2-2B*, and *GmFAD2-2C*—and created a CRISPR/Cas9-mediated single gene editing vector system. The results of Sanger sequencing showed that 72 transformed plants positive for T_1_ generation were obtained using *Agrobacterium*-mediated transformation, of which 43 were correctly edited plants, with the highest editing efficiency of 88% for *GmFAD2-2A*. The phenotypic analysis revealed that the oleic acid content of the progeny of *GmFAD2-1A* gene-edited plants had a higher increase of 91.49% when compared to the control JN18, and the rest of the gene-edited plants in order were *GmFAD2-2A*, *GmFAD2-1B*, *GmFAD2-2C*, and *GmFAD2-2B*. The analysis of gene editing type has indicated that base deletions greater than 2bp were the predominant editing type in all editing events. This study provides ideas for the optimization of CRISPR/Cas9 gene editing technology and the development of new tools for precise base editing in the future.

## 1. Introduction

Soybean is one of the important oil crops in China, providing major vegetable protein for human food, animal feed, and industrial use. Soybean contains 17–22% of seed oil, is also widely used for industrial applications and bio-diesel production [1,2], and has an important role in food security, ecological security, and sustainable and stable agricultural development in China [3]. The soybean is a self-pollinated crop with a natural heterosis rate of 0.5–1.0% [4], and there are many limitations in traditional breeding to improve the oil and oleic acid content of soybean seeds [5]. Therefore, the application of genetic engineering to improve the oil content in soybean seeds based on the understanding of the molecular mechanisms regulating plant oil metabolism is of great importance for food supply, industrial production, and new energy utilization [6,7].

CRISPR/Cas9 gene editing technology is a hot tool for molecular breeding in recent years [8], which is low cost, is more accurate, is easy to use, and allows targeted gene manipulation at multiple locations throughout the genome and simultaneous editing to achieve efficient genome editing in various plants [9]. However, it still faces limitations such as off-target and low gene editing efficiency [10,11]. In recent years, scientists have used different promoter assemblies to drive Cas9 or multiple gRNAs in editing vectors to guide Cas9 to target multiple genes to optimize the effect of gene editing breeding [12]. However, in different studies reported we found that the pattern of gene editing occurring with Cas9 was different, and the achieved effect was also different, which shows that the pattern of gene editing occurred is an essential factor affecting the breeding effect [13]. In 2018, Ryosuke et al. generated multiple mutation types by designing different promoters to drive Cas9 in pMgPsef1-gRNA1/2#1 and #4 lines with more than 1000 bp base deletions with low-level chimeras, achieving good breeding results [14]. In 2019, Phat T. et al. used double gRNA to simultaneously knock out *FAD2-1A* and *FAD2-1B* to create a double allele mutant high oleic acid low linoleic acid mutant resource with editing types including deletions, insertions, and inversions [15]. In 2020, Chen et al. designed a double allele editing vector for editing *GhFAD2-1A* and *GhFAD2-1D* based on CRISPR/Cas9 gene editing technology, and 69.57% of the 19 positive plants with correct editing were of the base deletion type. Most of them were “C“ deletion (86.84%). The oleic acid content of M20-2 seeds that were successfully edited by fatty acid analysis was 5.58 times higher than the wild type, and the linoleic acid content decreased from 58.62% to 6.85% [16]. In 2021, Naoufal Lakhssassi et al. used EMS mutagenesis to identify multiple types of mutations in the *GmFAD2* gene, with 74% of the substitutions from G to A and from C to T [17]. In 2021, Cai et al. designed two sgRNAs to target and edit the *GmJAG1* gene, and sequencing results showed that base deletions in gene-editing positive plants resulted in the knockout of the start codon of *GmJAG1* and *GmJAG2* and partial deletion in the second exon [18]. In 2021, Wang et al. successfully created double and quintuple mutants with 1 to 60 single base insertions or deletions, resulting in changes in the amino acid sequence of the corresponding GmAITR protein, by designing multiple gene editing vectors. They found by sequencing that the *GmFAD2-1A* gene had a G to A mutation and resulted in a change in the tryptophan codon (TGG) at amino acid 293 to a premature stop codon, leading to an increase in oleic acid content [19]. Different types of mutation patterns produce different effects, and most of the key mutations and agronomically important genetic variants are single base polymorphisms (single nucleotide polymorphisms) that require precise genome editing tools to correct the sequence [20,21].

Utilizing the CRISPR/Cas9 gene editing system, we knocked out five key enzyme genes of the *GmFAD2* gene family and compared the phenotypic changes in gene editing-positive soybean that produced different types of editing, to identify the optimal germplasm traits. This study aims to optimize the application of the CRISPR/Cas9 gene editing system in molecular breeding and base editing.

## 2. Results

### 2.1. Construction of Gene-Editing Vectors

The CRISPR-P online program (http://crispr.hzau.edu.cn/CRISPR2/) was used to design gRNA target sequences for *GmFAD2-1A*, *GmFAD2-1B*, *GmFAD2-2A*, *GmFAD2-2B*, and *GmFAD2-2C* based on sequence specificity and high GC content in exon sequences. The gRNA target sequences are listed in Table 1. Five single gene editing vectors were constructed for *GmFAD2-1A*, *GmFAD2-1B*, *GmFAD2-2A*, *GmFAD2-2B*, and *GmFAD2-2C* using the Hangzhou Baige (Hangzhou, China) CRISPR/Cas9 vector BGK015. Sequencing of the PCR amplicons indicated that the target gene size and sequence were as expected, which confirmed that the single gene editing vectors were successfully constructed (Appendix A).

### 2.2. Molecular Analysis of the Transgenic Progeny

Analysis was conducted after the creation of gene editing positive plants according to the breeding process in Figure 1. Genetic transformation of soybean was obtained by using the *Agrobacterium*-mediated method (Appendix A). PCR amplification of the herbicide resistance marker genes *Bar* and Cas9 identified 72 T_1_ transgenic plants, which included 17 transgenic plants of *GmFAD2-1A* (designated JN18-CR1AT_1_-1 to JN18-CR1AT_1_-17), 15 of *GmFAD2-1B* (designated JN18-CR1BT_1_-1 to JN18-CR1BT_1_-15), 18 of *GmFAD2-2A* (designated JN18-CR2AT_1_-1 to JN18-CR2AT_1_-18), 14 of *GmFAD2-2B* (designated JN18-CR2BT_1_-1 to JN18-CR2BT_1_-14), and 8 of *GmFAD2-2C* (designated JN18-CR2CT_1_-1 to JN18-CR2CT_1_-8) (Figure 2 and Figure 3). Following the sowing of the T_1_ transgenic plant seeds, the resulting plants were grown to maturity and later harvested. Subsequently, the T_2_ generation lines were procured. The partial PCR amplification results are shown in Figure 4 (Appendix A).

To determine whether the exogenous gene was integrated into the genomes of the T_2_ plants, Cas9 was used as a probe to perform Southern blotting detection on some of the T_2_ generation editing plants. The results showed that the non-transformed negative plants had no hybridization signals, while the editing plants showed that foreign genes had been integrated into the genome in the form of a single copy, and the hybridization signals between individuals were slightly different (Figure 5).

### 2.3. Sequence Analysis of the Target Gene Mutations in T_1_ Transgenic Plants

In this study, 72 transgene-positive plants were identified by PCR amplification of T_1_ generation transgenic plants using target-specific primers. Sanger sequencing was performed and the Cc-qPCR method reported in the study was used to determine the mutation types in the gene-editing positive plants and to count the editing efficiency [22]. A total of 43 gene editing positive plants with correct editing events were obtained based on sequencing results and analysis of qPCR results. Of the 18 independent positive plants obtained for *GmFAD2-2A*, 16 had mutated sequences (mutation efficiency of 88.9%), of which 10 were homozygous mutants, and 6 were heterozygous mutants. The 14 independent positive plants obtained for *GmFAD2-2B* included 9 plants with mutant sequences (mutation efficiency of 64.2%), of which 6 were homozygous mutants, and 3 were heterozygous. Of the 17 independent positive plants obtained for *GmFAD2-1A*, 7 had mutant sequences (mutation efficiency of 41.1%), 3 were homozygous mutants, and 4 were heterozygous. A total of 15 independent positive plants were obtained for *GmFAD2-1B*, including 6 plants with mutant sequences (mutation efficiency of 40%), of which 3 were homozygous, and 3 were heterozygous. The eight independent positive plants obtained for *GmFAD2-2C* included five plants with mutant sequences (mutation efficiency of 62.5%), of which three were homozygous mutants and two were heterozygous mutants. Part of the sequencing results and graphical analysis results are shown in Figure 6 and Figure 7.

Analysis of the mutation types in the target sequences revealed eight types, including base deletions greater than 2 bp, single base G substitutions, single base A substitutions, single base T substitutions, single base A insertions, single base G insertions, single base T deletions, and single base G deletions (Figure 7). Subsequent amino acid sequence comparisons and raw signal analysis showed that the mutants that produced the correct editing events all had significantly altered amino acid sequences and the protein structures were predicted to be significantly different (Figure 8). Mutant individuals exhibited early termination of transcription, with significant differences in secondary and tertiary structure predictions. Most mutations were localized to key protein structural domains, including enzyme catalytic activity, homodimer interfaces, or substrate binding, suggesting that isolated mutations may negatively affect protein activity or dimerization (Appendix A).

Of all plants with correct editing events, those with base deletions greater than 2 bp had the highest average editing efficiency of 48%. Single base T deletions had an average editing efficiency of 18%. Single base G deletions had an average editing efficiency of 11%. Single base R substitutions had an editing efficiency of 7%. Single base A insertions had an editing efficiency of 5%. Single base A substitutions had an editing efficiency of 4%. Single base G insertions had an editing efficiency of 3%. Single base T substitution editing efficiency was 2% (Figure 9).

### 2.4. Phenotypic Identification of T_2_ Transgenic Soybean Seed

#### 2.4.1. Identification of T_2_ Transgenic Soybean High Expression Strains

Quantitative real-time PCR analysis was performed using a β-actin-encoding gene (LOC100798523) as a reference control. The generated data showed that compared to the control (JN18), the three lines with the most significant relative expression decrease for each target gene were chosen for phenotypic identification and further functional analysis of T_2_ transgenic plants (Figure 10).

#### 2.4.2. Determination of the Major Fatty Acid Content of T_2_ Transgenic Soybeans

The fatty acid content of seeds from some T_2_ transgenic soybean lines was analyzed. In contrast to the seeds of the control Jinong 18, the oleic acid content of the five transgenic lines of editing vectors was significantly higher, with the *GmFAD2-1A* gene having the greatest increase of 91.49%, then *GmFAD2-1B* (89.21%), *GmFAD2-2A* (82.31%), *GmFAD2-2C* (78.87%), and *GmFAD2-2B* (75.70%). On the contrary, the linoleic acid content of all lines that tested positive for the editing vector was found to be lower by 7.1–10.3% when compared to the control, while no significant differences were observed for other quality traits (Figure 11, Table 2).

Further investigation indicated that the oleic acid content in the seeds of transgenic lines with different editing types of vectors was substantially increased compared to the control JN18. The average increase of editing lines with base deletion greater than 2 bp was up to 105.51%. The oleic acid content of seeds with single base G substitution increased by 95.77% on average. The oleic acid content of seeds with single base A substitution increased by 93.66% on average. The oleic acid content of seeds with single base T substitution increased by 90.16% on average. The oleic acid content of seeds with single base A insertion increased by 84.87%. Seed oleic acid content increased by 79.92% for single base G insertion. Seed oleic acid content increased by 75.81% for single base T deletion, and seed oleic acid content increased by 74.25% for single base G deletion. Contrarily, we observed a 7.5–14.0% reduction in linoleic acid content in all gene editing positive plants in the different editing types compared to the control, while no significant differences were observed for other quality traits (Figure 12, Table 3). The research results confirm that *GmFAD2*, a soybean fatty acid dehydrogenase, is the essential enzyme for regulating oleic acid to linoleic acid.

#### 2.4.3. Determination of Major Agronomic Traits in T_2_ Transgenic Soybean

The analysis results of main agronomic traits indicated that in comparison to the control JN18, there were no major differences in other major traits such as flower color, hairy, leaf type, plant height, the number of effective branches, and nodes between different editing vectors; however, there were three phenotypic traits related to yield traits, such as the number of pods per plant, the total number of grains, and the weight of the grains per plant, that showed significant differences. Moreover, analysis results of different editing types showed similarity to those of editing vectors, with the exception of significant differences in pod number per plant, grain number per plant, and grain weight per plant when compared to the control. There were not significant differences in other major agronomic traits (Table 4 and Table 5).

## 3. Discussion

In this study, we obtained gene editing positive plants by the transformation of soybean through the *Agrobacterium*-mediated method, in which we found the highest editing efficiency of *GmFAD2-1A* among the five key enzyme genes, reaching 88%. Compared with previous reports on soybean and peanut, the editing efficiency was improved by 10–20% [15,16,17]. In plant molecular breeding, the gene editing efficiency of CRISPR/Cas9 is an essential factor affecting the effectiveness of molecular breeding, and improving the editing efficiency of target genes is the basis for creating more excellent traits in plants.

To explore the different effects of different gene editing models on the phenotypes of crops, we selected five essential enzyme genes of the *FAD2* gene of the soybean fatty acid dehydrogenase family, which have been widely reported in previous studies, and used the most popular tool for gene editing breeding, the CRIPSR/Cas9 gene editing system, to explore and compare the editing options with the most significant impact on the phenotypes of agronomic traits [22,23]. The genetic principle of CRISPR/Cas9 is to use Cas9 to induce DSB at target sites on a plant-specific genome under gRNA guidance [24,25]. With insertions or deletions, and although NHEJ-mediated mutagenesis is highly effective in plants, it is commonly used to generate knockouts and alter promoter or enhancer strengths necessary to achieve precise genome editing to develop new agronomic traits [26]. Although the new base editing technology BE was reported in 2016 and has improved the efficiency and the accuracy of base editing and major editing compared to the traditional CRISPR/Cas9 technology, the impact of targeting different target sites and the different approaches that occur on gene editing efficiency and editing effectiveness is still significant and is a major problem to be addressed in future plant gene editing [27,28]. In 2017, Naoufal Lakhssassi et al. used mutagenesis to identify one C to G, three C to T mutations in the *GmFAD2-1A* gene, and one C to T mutation in *GmFAD2-1B*, resulting in a 30% to 50% increase in oleic acid content, similar to the effect achieved with our gene editing vector, where replacement mutations did not exist [29]. In 2020, Nan et al. identified a G substitution in *FAD2-1A* that caused a Trp mutation at position 254 to become a stop codon, increasing in oleic acid content [30]. In 2020, Wu et al. constructed two gene editing and one gene editing vectors for the *GmFAD2-1A* and *GmFAD2-1B* genes. Two gene editing and one double gene editing vectors were constructed, in which a single gene editing vector with a 2bp deletion and a base T deletion occurred, the oleic acid content was increased by 87.55% and 141.5%, and a double gene editing vector with both a 7bp deletion and a 2 deletion at both targets increased the oleic acid content by 329.3%, showing that two essential enzyme genes with simultaneous large block deletions can exert a powerful gene editing for breeding [31]. Efficient gene editing is also one of the key factors that affect breeding performance, and stable genetic transformation of soybeans remains one of the most difficult challenges to overcome [32].

In recent years, scientists have been optimizing the CRISPR/Cas9 system to improve its editing efficiency [33]. In 2019, Ren et al. compared the effects of sgRNA-CG content and SpCas9 expression levels on gene editing efficiency to optimize CRISPR/Cas9 editing efficiency. In 2020, Zheng et al. used the ubiquitin-related structural domain (UBA) to enhance Cas9 protein stability to improve editing efficiency [34]. In 2020, multiple gRNA-CRISPR/Cas9 vectors were constructed using multiple cis-trans tRNA-gRNA approaches targeting the Medicago *Sativastay-green* (MsSGR) gene. Replacement of the CaMV35S promoter with the anthocyanin promoter (AtUBQ10) to drive Cas9 expression in the multiple gRNA systems resulted in a significant increase in genome editing efficiency [35]. In 2022, Zhang et al. found that the MaU6c promoter was approximately four times more active than the OsU6a promoter in banana protoplasts, and the application of this promoter in CRISPR/Cas9 and banana codon-optimized Cas9 resulted in a fourfold increase in mutational efficiency compared to the previous banana CRISPR/Cas9 [36]. In 2022, Patrick et al. engineered an improved temperature-tolerant variant of Cas12a from the bacterium *Lachnospiraceae* ttLbCas12a, which at a standard incubation temperature of 22 °C in Arabidopsis showed significantly improved editing efficiency over LbCas12a [37,38,39]. There are relatively few reports on the optimization of the CRISPR/Cas9 technology in soybeans [40]. This study further optimizes CRISPR/Cas9 technology by comparing the effects of gene editing modes on phenotypes, which will further facilitate basic molecular research and molecular breeding techniques in various plant species, including useful crops, and is one of the most useful genome editing tools in plant genome engineering. We explore the effects of different editing modes. The impact of editing modes on agronomic traits will lay the foundation for future optimization of the optimized CRISPR/Cas9 technology, the development of base editors, and their widespread use in molecular breeding [41,42,43,44,45]. This study explores the different effects of different editing types on plant phenotypes by comparing the changes in the oleic acid content of transgenic soybeans undergoing different editing patterns, and develops ideas for future precise base editing.

## 4. Materials and Methods

### 4.1. Materials

The “Jinong 18” soybean variety (Jishendou 2006 was carefully chosen from the foreign-sourced initial generation lines by the Agricultural College of Jilin Agricultural University in 2006, and it is a high-oil soybean variety) selected as recipient material in this study was provided by the Jilin Provincial Key Laboratory of Plant Molecular Breeding.

### 4.2. Construction of CRISPR/Cas9 Editing Vectors

High-scoring gRNA sequences were selected from the CRISPR-P website at http://cbi.hzau.edu.cn/cgi-bin/CRISPR (accessed on 22 January 2021), and in vitro activity was tested to identify gRNAs with >90% activity. The gRNA targets designed above were sequenced on the website at http://www.biogle.cn/ (accessed on 14 May 2021). Oligo sequences were generated from the kit by dissolving the synthesized oligo in water at 10 µM, mixing buffer anneal 18 µL, UP oligo 1 µL, and low oligo 1 µL; adding ddH_2_O to a total volume of 20 µL; heating at 95 °C for 3 min; and then slowly reducing to 20 °C at approximately 0.2 °C/S. The components were mixed on ice according to CRISPR/Cas vector 2 μL, oligo dimer 2 μL, enzyme mix 1 μL, plus ddH_2_O to a total volume of 10 μL, mixed and reacted at room temperature (20 °C) for 1 h, and transformed into *E*. *coli*. The DNA from the recombinant plasmid was extracted for PCR validation, and 50 μL of plasmid DNA was sequenced by Sanger and subsequently compared by DNAMAN software (V6.0) to verify whether the gRNA was successfully ligated into the vector (Appendix A).

### 4.3. Soybean Genetic Transformation and PCR Analysis of the Progeny of the Transgenic Plants

The cotyledon node of soybean “JN18” was utilized as the receptor material, and the five constructed CRISPR/Cas9 editing vectors were transferred into the receptor by *Agrobacterium* tumefaciens-mediated method, thereby obtaining the transgenic soybean plants. Genomic DNA from the T_0_ soybean plants was amplified by PCR using primers Cas9-1F/Cas9-1R and Bar-1F/Bar-1R (Appendix A). The PCR amplifications were completed in a final reaction volume of 20 μL. The PCR conditions for amplifying the Bar gene were as follows: 94 °C for 5 min; 30 cycles of 94 °C for 30 s, 60 °C for 30 s, and 72 °C for 1 min. The PCR conditions for amplifying the Cas9 gene were as follows: 94 °C for 5 min; 30 cycles of 94 °C for 30 s, 55 °C for 30 s, and 72 °C for 1 min; 72 °C for 10 min (Appendix A).

### 4.4. Evaluation of Transformation Efficiency and Gene Editing Efficiency

Specific primers were designed to include the target site. Genomic DNA was extracted from the positive plants for PCR amplification. The amplicons were sequenced by Changchun Kumei (Changchun, China). The generated sequences were compared and analyzed using DNAMAN (v6.0). The base changes within the gRNA sequence indicated that the target gene was successfully edited. Statistics on the efficiency of transformation and gene editing in soybeans were obtained.

The amino acid sequences of the transgenic positive plants obtained by Sanger sequencing were used as templates. The SWISS-MODEL website (http://swissmodel.expasy.org/, accessed on 19 October 2021) was opened, and the amino acid sequences were submitted for template identification.

### 4.5. Southern Blot Analysis of the T_2_ Transgenic Plants

To confirm whether the exogenous gene has been integrated into the positive plant genome, genomic DNA from the transgenic plant leaves was extracted and double-digested with restriction endonucleases *Hind* III and *BamH* I. Utilizing an exogenous gene Cas9 as a probe, Southern blot analysis was conducted with the DIG-HIGH Prime DNA Labeling and Detection Starter Kit I (C58-11745832910, Roche company, Shanghai, China).

### 4.6. Quantitative Real-Time Polymerase Chain Reaction Analysis (q-RT PCR)

Total RNA extracted from confirmed T_2_ transgenic plants was used as the template for synthesizing cDNA. The q-RT PCR analysis was completed in a final reaction volume of 20 μL, which included 10 µL TB Green Premix Ex Taq (Tli RNaseH Plus), 0.4 μL PCR forward primer (10 μM), 0.4 μL PCR reverse primer (10 μM), 0.4 μL ROX Reference Dye (50×), 2 μL cDNA template, and 6.8 μL RNase-free water. The q-RT PCR analysis was performed using the Agilent MX3000P PCR instrument, with the following program: 95 °C for 3 min; 40 cycles of 95 °C for 5 s and 60 °C for 30 s (Appendix A). This was followed by a melting curve analysis involving the following program: 95 °C for 15 s, 60 °C for 1 min, and 60 °C for 30 s. Relative gene expression values were calculated using the 2^−ΔΔCt^ method, and analysis of plant mutation types by Cc-qPCR [22]. Three separate biological replicates were performed with each treatment. Variance analysis was performed using GraphPad Prism (8.0).

### 4.7. Determining the Primary Characteristics of Quality

The NIRS DS2500 NIR quality analyzer (FOSS, Stockholm, Sweden) was used to determine the protein, oil, and major fatty acid content of mature soybean seeds, selecting full, insect-free, uniformly sized seeds to be placed in the measuring cup, laying them flat, and placing them in the NIR sample tank. The number of seeds needed to fill the measuring cup (approx. 60 g) is covered over the infrared sensing area and collected by the operating software Operator according to the collection library previously established in the laboratory. Each measurement setting was repeated three times, and the results were automatically saved to a computer. The data were analyzed by IBM SPSS Statistics (27.0.1).

### 4.8. Analysis of the Main Agronomic Traits

The positive seeds obtained from the T_1_ generation were sown in the experimental field of Jilin Agricultural University, with each strain planted in a row of 4.5 m in length, 10 cm between plants, and 65 cm between rows, double-spaced, and managed in the field under natural rainfall conditions. When the plants were fully mature, three plants of each line were randomly selected for indoor testing of the main agronomic traits.

The agronomic traits of the soybean variety were assessed by the “China Soybean Variety Records”. The criteria for yield traits included the number of pods per plant, the grain weight per plant, and the 100-grain weight. The number of pods was determined by measuring an average of three plants per line. The grain weight was determined by measuring the average of three plants, and the 100-grain weight was determined by measuring the weight of 100 grains of uniform size from a single plant, repeated three times (g). Other agronomic traits assessed included plant height, the number of branches, and the number of nodes. Plant height was measured from the cotyledon node to the top of the main stem in the laboratory (cm). The number of branches and nodes on the main stem was counted. The data were analyzed by IBM SPSS Statistics (27.0.1).

### 4.9. Statistical Analysis

The data are expressed as the mean ± SD of the values obtained in the repeated experiments. For statistical analysis, variance analysis was performed using IBM SPSS Statistics (27.0.1). Tukey ‘s multiple comparison tests were performed, and then multiple comparison tests and calculations of LSD were performed. Differences with *p* values less than 0.05 were considered statistically significant.

## 5. Conclusions

In summary, we constructed five *GmFAD2* gene editing vectors, created mutant strains with 62.15–100.65% higher oleic acid content than the control JN18, and achieved 88% editing efficiency of the *GmFAD2-1A* gene. Comparing the effects of different editing patterns in the *GmFAD2* gene family on the oleic acid content of positive transgenic plants, we found that mutations with base deletion types greater than 2 bp resulted in large changes in amino acid sequence caused by the premature functioning of the stop codon, which affected the production of linoleic acid catalyzed by soybean fatty acid dehydrogenase, resulting in elevated oleic acid content in seeds. The knockout of large segments in the coding region of the target gene tended to produce more pronounced breeding effects. Our research opens up new ideas for further development of gene editing breeding technologies.

## Figures and Tables

**Figure 1 ijms-24-04769-f001:**
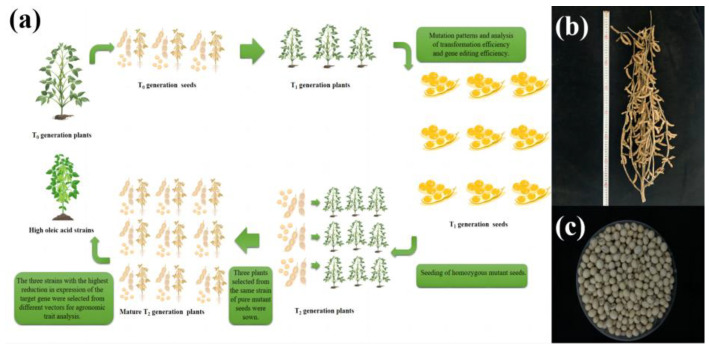
Overview of the gene editing plant creation process and phenotype: (**a**) gene editing breeding process; (**b**) gene editing positive plants; (**c**) gene editing positive seeds.

**Figure 2 ijms-24-04769-f002:**

PCR detection of Cas9 in some T_1_ generation transgenic plants: (**A**) 1–3 transgenic positive plants; (**B**) 1–3 transgenic positive plants; (**C**) 1–4 transgenic positive plants; (**D**) 1–7 transgenic positive plants; (**E**) 1–4 transgenic positive plants; (**F**) 1–4 transgenic positive plants. M: DL2000 DNA marker; +: positive plasmid; -: negative control.

**Figure 3 ijms-24-04769-f003:**
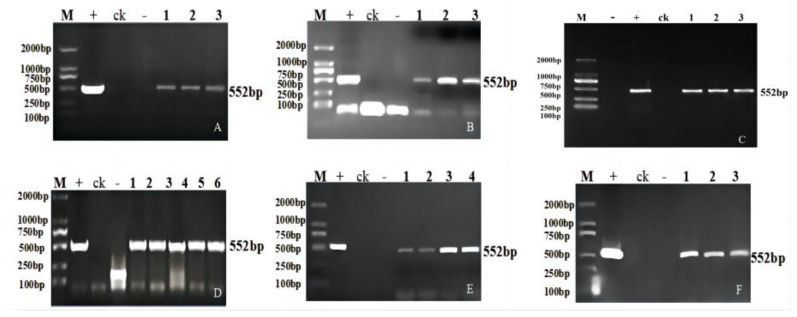
PCR detection of *Bar* gene in some T_1_ generation transgenic plants: (**A**) 1–3 transgenic positive plants; (**B**) 1–3 transgenic positive plants; (**C**) 1–3 transgenic positive plants; (**D**) 1–7 transgenic positive plants; (**E**) 1–4 transgenic positive plants; (**F**) 1–3 transgenic positive plants. M: DL2000 DNA marker; +: positive plasmid; -: negative control.

**Figure 4 ijms-24-04769-f004:**
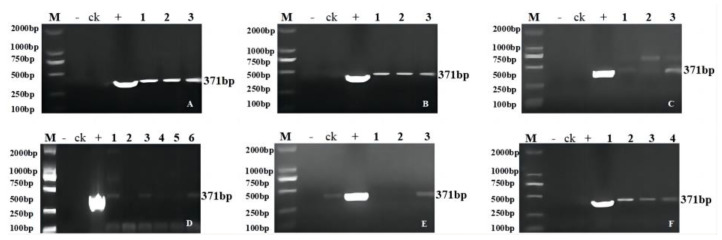
PCR detection of Cas9 in some T_2_ generation transgenic plants: (**A**) 1–3 transgenic positive plants; (**B**) 1–3 transgenic positive plants; (**C**) 1–3 transgenic positive plants; (**D**) 1–6 transgenic positive plants; (**E**) 1–3 transgenic positive plants; (**F**) 1–4 transgenic positive plants. M: DL2000 DNA marker; +: positive plasmid; -: negative control.

**Figure 5 ijms-24-04769-f005:**
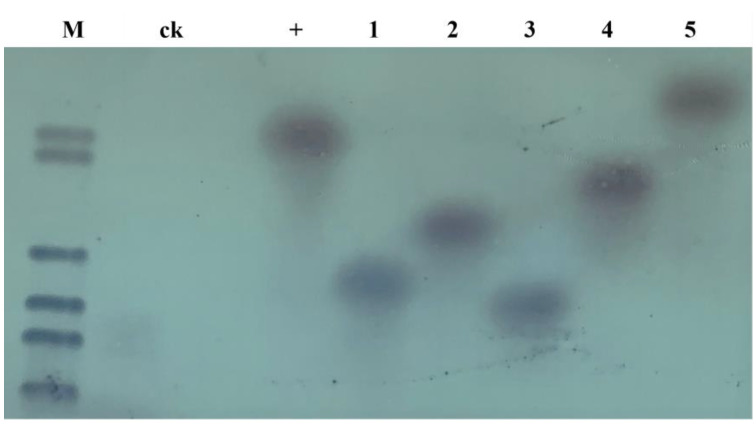
Southern blot detection of some T_2_ transgenic plants: M: DNA marker +: positive plasmid; CK: untransformed plant; 1: transfer *GmFAD2-2A* on soybean; 2: transfer *GmFAD2-2B* on soybean; 3: transfer *GmFAD2-1A* on soybean; 4: transfer *GmFAD2-1B* on soybean; 5: transfer *GmFAD2-2C* on soybean.

**Figure 6 ijms-24-04769-f006:**
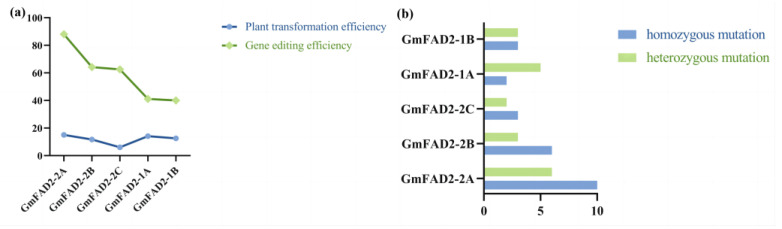
Editing efficiency and mutation type of different editing vectors: (**a**) comparison of editing efficiency and conversion efficiency of different editing vectors; (**b**) edited plant mutation types.

**Figure 7 ijms-24-04769-f007:**
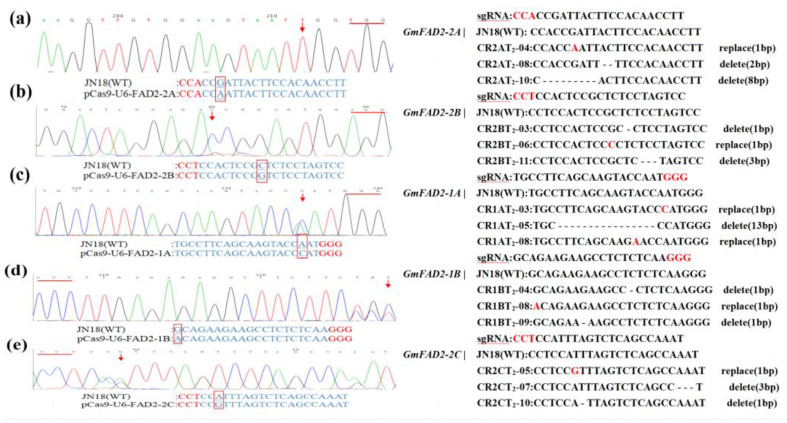
Part of Sanger sequencing results and *GmFAD2* gene sequence mutation sites: (**a**) pBGK015Cas9-U6-FAD2-2A; (**b**) pBGK015Cas9-U6-FAD2-2B; (**c**) pBGK015Cas9-U6-FAD2-1A; (**d**) pBGK015Cas9-U6-FAD2-1B; (**e**) pBGK015Cas9-U6-FAD2-2C.

**Figure 8 ijms-24-04769-f008:**
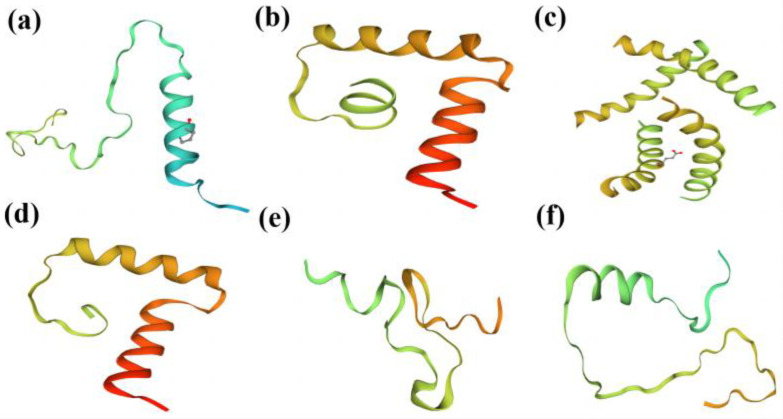
Partial prediction diagram of protein tertiary structure: (**a**) wild; (**b**) FAD2-1A mutant; (**c**) FAD2-1B mutant; (**d**) FAD2-2A mutant; (**e**) FAD2-2B mutant; (**f**) FAD2-2C mutant.

**Figure 9 ijms-24-04769-f009:**
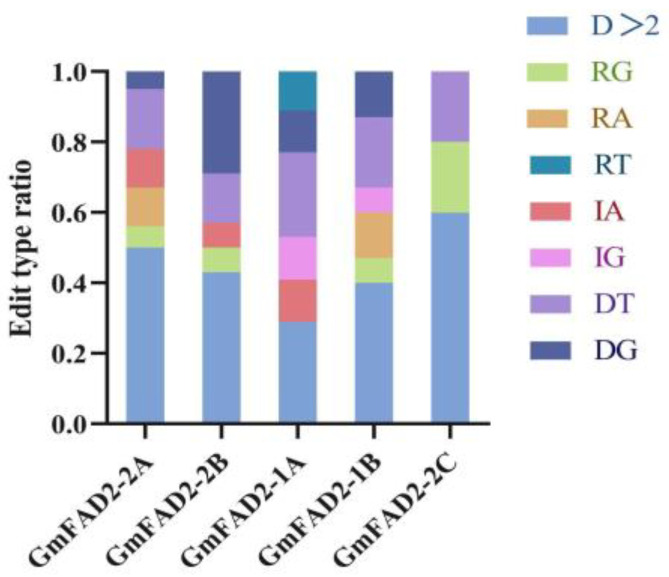
Proportion of different editing modes in different editing carriers.

**Figure 10 ijms-24-04769-f010:**
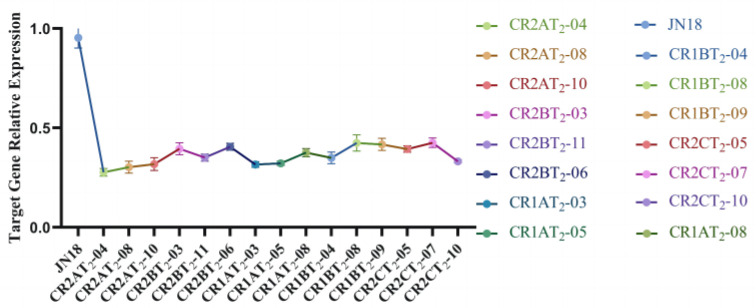
Expression of target gene in gene editing positive plants. All data shown are means ± SD of three biological replicates.

**Figure 11 ijms-24-04769-f011:**
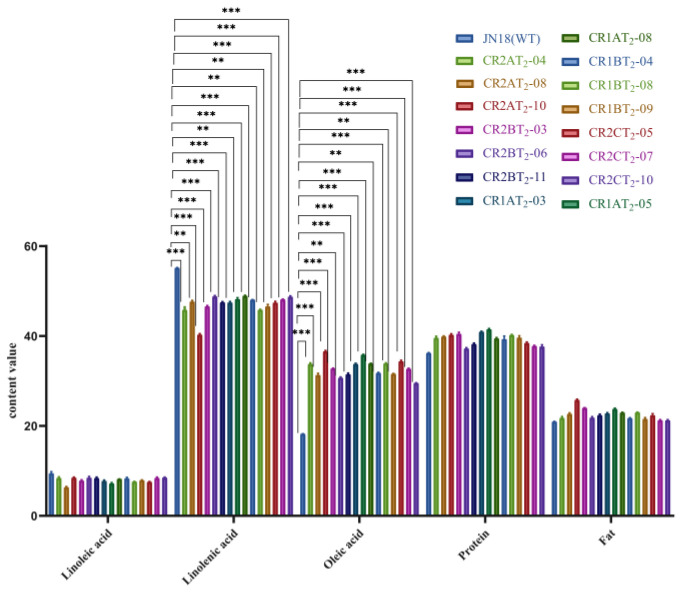
Comparison of quality traits of different mutation types. Significant differences according to Student’s t-test are indicated. **, and *** indicated significant differences at *p* < 0.01, and *p* < 0.001 levels, respectively.

**Figure 12 ijms-24-04769-f012:**
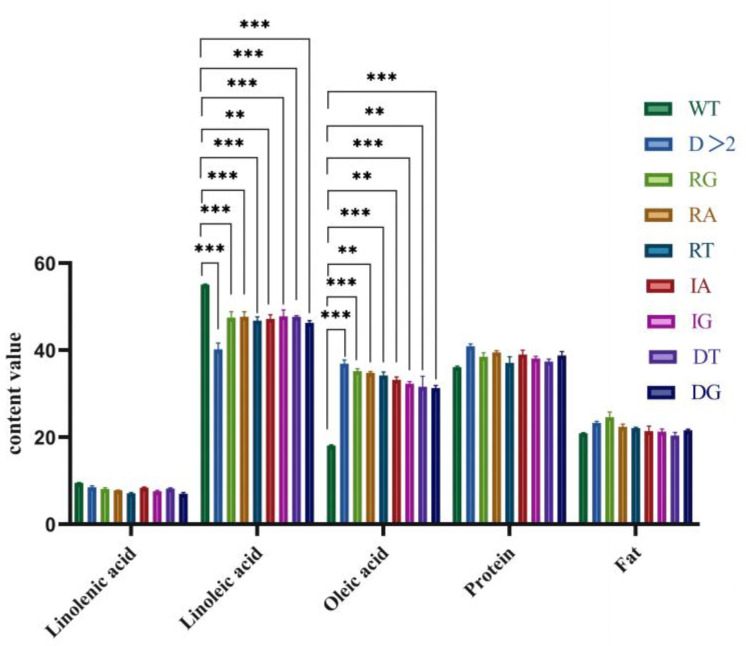
Comparison of quality traits of gene editing positive plants. Significant differences according to Student’s *t*-test are indicated. ** and *** indicated significant differences at *p* < 0.01, and *p* < 0.001 levels, respectively. All data shown are means ± SD of three biological replicates.

**Table 1 ijms-24-04769-t001:** gRNA-guided sequence of target gene.

Name	Target Sequence	GC Content (%)	The Target Rate
FAD2-2A-Target	AAGGTTGTGGAAGTAATCGGTGG	45	0.5466
FAD2-2B-Target	GGACTAGGAGAGCGGAGTGGAGG	65	0.6594
FAD2-1A-Target	TGCCTTCAGCAAGTACCAATGGG	45	0.6309
FAD2-1B-Target	GCAGAAGAAGCCTCTCTCAAGGG	50	0.7301
FAD2-2C-Target	ATTTGGCTGAGACTAAATGGAGG	40	0.6271

**Table 2 ijms-24-04769-t002:** Determination of fatty acid content of different editing vectors in T_2_ plants.

Materials	Oleic Acid (%)	Linoleic Acid (%)	Linolenic Acid (%)	Protein (%)	Fat (%)	Oleic Acid Increase Percentage (%)
JN18-CK	17.98 ^c^ ± 1.16	55.25 ^a^ ± 0.58	8.17 ^a^ ± 0.29	38.50 ^b^ ± 0.83	21.85 ^b^ ± 1.67	-
CR1AT_2_	34.43 ^a^ ± 1.29	46.30 ^bc^ ± 0.68	7.78 ^a^ ± 0.52	40.16 ^a^ ± 0.99	23.14 ^ab^ ± 0.55	91.49
CR1BT_2_	34.02 ^a^ ± 1.61	46.87 ^b^ ± 1.14	7.96 ^a^ ± 0.40	39.14 ^b^ ± 0.67	22.11 ^b^ ± 0.81	89.21
CR2AT_2_	32.78 ^ab^ ± 3.29	44.92 ^c^ ± 1.29	7.89 ^a^ ± 1.16	39.56 ^ab^ ± 0.36	23.41 ^a^ ± 2.11	82.31
CR2BT_2_	31.59 ^bc^ ± 1.03	47.62 ^bc^ ± 1.12	8.22 ^a^ ± 0.35	38.59 ^b^ ± 1.69	22.73 ^ab^ ± 1.10	75.70
CR2CT_2_	32.16 ^b^ ± 2.40	48.15 ^b^± 0.73	8.13 ^a^ ± 0.52	38.42 ^b^ ± 0.39	21.57 ^b^ ± 0.58	78.87
LSD_0.05_	1.61	1.88	0.49	0.76	0.95	-

Note: Data are expressed as mean ± standard deviation (X ± SD); LSD_0.05_ indicates least significant difference values calculated at the α = 0.05 level. a, b, and c was used to indicate the significant level of *p* = 0.05.

**Table 3 ijms-24-04769-t003:** Determination of fatty acid content of different edit types in T_2_ plants.

Type of Editing	Oleic Acid (%)	Linoleic Acid (%)	Linolenic Acid (%)	Protein (%)	Fat (%)	Oleic Acid Increase Percentage (%)
JN18-CK	17.98 ^d^ ± 1.16	55.25 ^a^ ± 0.58	8.17 ^ab^ ± 0.29	38.50 ^b^ ± 0.83	21.85 ^b^ ± 1.67	-
D > 2	36.95 ^a^ ± 0.85	41.22 ^d^ ± 1.74	8.55 ^a^ ± 0.33	39.92 ^a^ ± 0.57	22.30 ^a^ ± 1.46	105.51%
RA	34.82 ^bc^ ± 0.32	47.17 ^bc^ ± 0.99	7.92 ^b^ ± 0.51	39.52 ^ab^ ± 0.39	21.61 ^b^ ± 0.74	93.66%
RT	34.19 ^c^ ± 0.82	46.82 ^b^ ± 0.85	7.81 ^b^ ± 0.12	38.37 ^b^ ± 1.47	21.03 ^b^ ± 0.77	90.16%
RG	35.20 ^b^ ± 0.56	47.53 ^b^ ± 1.34	8.12 ^b^ ± 0.32	38.51 ^b^ ± 0.93	22.47 ^ab^ ± 3.06	95.77%
DT	31.61 ^d^ ± 0.48	47.67 ^b^ ± 0.29	8.20 ^ab^ ± 0.17	38.42 ^b^ ± 0.62	22.94 ^ab^ ± 1.06	75.81%
DG	31.33 ^d^ ± 0.60	46.33 ^c^ ± 0.49	7.72 ^b^ ± 0.33	38.76 ^b^ ± 0.99	23.03 ^a^ ± 0.69	74.25%
IA	33.24 ^d^ ± 0.64	47.65 ^b^ ± 1.23	8.40 ^ab^ ± 0.71	39.00 ^b^ ± 1.02	21.71 ^b^ ± 0.82	84.87%
IG	32.35 ^d^ ± 0.48	47.79 ^b^ ± 1.50	7.89 ^b^ ± 0.24	38.45 ^b^ ± 0.52	21.43 ^b^ ± 1.42	79.92%
LSD_0.05_	0.84	0.92	0.41	0.71	1.19	-

Note: D > 2 indicates a base deletion greater than 2 bp; RT indicates a single base T substitution; RA indicates a single base A substitution; RG indicates a single base R substitution; DT indicates a single base T deletion; DG indicates a single base G deletion. Data are expressed as mean ± standard deviation (X ± SD); LSD_0.05_ indicates least significant difference values calculated at the α = 0.05 level. a, b, c, d was used to indicate the significant level of *p* = 0.05.

**Table 4 ijms-24-04769-t004:** Determination of agronomic traits of different editing vectors in T_2_ plants.

Materials	Plant Height (cm)	Number of Branches	Number of Nodes	Number of Pods	Total Number of Grains	Single Plant Weight (g)	100 Grain Weight (g)
JN18-CK	83.28 ^a^ ± 1.89	5.92 ^ab^ ± 0.33	18.60 ^b^ ± 0.78	35.85 ^c^ ± 4.69	245.86 ^c^ ± 3.01	41.45 ^c^ ± 1.19	20.98 ^ab^ ± 0.48
CR1AT_2_	84.30 ^a^ ± 2.19	6.38 ^a^ ± 1.40	21.50 ^a^ ± 1.28	47.44 ^b^ ± 1.99	306.57 ^ab^ ± 4.95	50.37 ^b^ ± 2.14	20.78 ^b^ ± 0.62
CR1BT_2_	85.13 ^a^ ± 1.39	6.27 ^ab^ ± 0.96	20.27 ^ab^ ± 0.71	50.83 ^a^ ± 3.69	300.10 ^b^ ± 10.13	54.90 ^a^ ± 1.18	21.35 ^ab^ ± 0.36
CR2AT_2_	83.60 ^a^ ± 3.20	6.52 ^a^ ± 2.55	22.07 ^a^ ± 2.18	53.57 ^a^ ± 4.51	310.40 ^a^ ± 16.37	51.59 ^b^ ± 5.10	21.47 ^a^ ± 0.65
CR2BT_2_	83.13 ^a^ ± 0.61	6.47 ^a^ ± 1.44	18.47 ^b^ ± 0.85	52.67 ^a^ ± 4.55	298.57 ^b^ ± 5.42	52.39 ^b^ ± 2.93	20.61 ^b^ ± 0.70
CR2CT_2_	83.18 ^a^ ± 1.81	5.70 ^b^ ± 0.46	19.11 ^b^ ± 0.64	52.23 ^a^ ± 2.61	303.23 ^ab^ ± 5.92	52.30 ^b^ ± 2.98	20.74 ^b^ ± 0.85
LSD_0.05_	2.01	0.67	1.85	2.77	7.32	2.36	0.51

Note: Data are expressed as mean ± standard deviation (X ± SD); LSD_0.05_ indicates least significant difference values calculated at the α = 0.05 level. a, b, and c were used to indicate the significant level of *p* = 0.05.

**Table 5 ijms-24-04769-t005:** Determination of agronomic traits of different edit types in T_2_ plants.

Materials	Plant Height (cm)	Number of Branches	Number of Nodes	Number of Pods	Total Number of Grains	Single Plant Weight (g)	100 Grain Weight (g)
JN18-CK	83.28 ^b^ ± 1.89	5.92 ^ab^ ± 0.33	18.60 ^b^ ± 0.78	35.85 ^e^ ± 4.69	245.86 ^e^ ± 3.01	41.45 ^b^ ± 1.19	20.98 ^b^ ± 0.48
D > 2	85.13 ^ab^ ± 2.85	6.03 ^ab^ ± 1.70	21.26 ^a^ ± 2.37	49.50 ^bc^ ± 5.48	315.40 ^a^ ± 7.99	51.85 ^a^ ± 4.89	21.15 ^b^ ± 0.61
RA	82.65 ^b^ ± 0.81	5.65 ^b^ ± 0.47	20.62 ^ab^ ± 1.03	46.03 ^c^ ± 3.21	298.51 ^bc^ ± 2.64	52.24 ^a^ ± 0.52	20.56 ^b^ ± 0.72
RT	83.53 ^b^ ± 1.40	5.85 ^ab^ ± 1.44	20.96 ^ab^ ± 1.99	40.95 ^d^ ± 1.89	290.93 ^c^ ± 18.06	44.15 ^b^ ± 4.09	19.61 ^c^ ± 0.57
RG	84.28 ^ab^ ± 2.81	5.50 ^b^ ± 0.72	19.86 ^ab^ ± 2.47	46.62 ^bc^ ± 1.86	279.80 ^d^ ± 17.91	50.31 ^a^ ± 3.74	22.07 ^a^ ± 0.61
DT	83.10 ^b^ ± 0.95	5.73 ^b^ ± 0.41	20.23 ^ab^ ± 1.51	53.73 ^a^ ± 1.06	299.86 ^bc^ ± 7.74	52.80 ^a^ ± 3.22	20.92 ^b^ ± 0.75
DG	85.41 ^ab^ ± 1.40	6.58 ^a^ ± 0.78	19.53 ^b^ ± 0.83	43.56 ^cd^ ± 8.55	307.12 ^ab^ ± 9.20	51.63 ^a^ ± 9.03	21.16 ^b^ ± 0.61
IA	85.62 ^a^ ± 3.83	4.99 ^b^ ± 0.75	19.57 ^b^ ± 2.46	49.56 ^b^ ± 1.07	297.46 ^bc^ ± 16.49	52.04 ^a^ ± 1.68	20.58 ^b^ ± 1.53
IG	85.70 ^a^ ± 2.74	5.39 ^b^ ± 1.13	20.25 ^ab^ ± 1.91	48.38 ^bc^ ± 1.42	302.78 ^b^ ± 4.37	44.23 ^b^ ± 0.94	20.51 ^b^ ± 0.89
LSD_0.05_	1.97	0.79	1.49	3.53	9.29	3.34	0.66

Note: D > 2 indicates a base deletion greater than 2 bp; RT indicates a single base T substitution; RA indicates a single base A substitution; RG indicates a single base R substitution; DT indicates a single base T deletion; DG indicates a single base G deletion. Data are expressed as mean ± standard deviation (X ± SD). LSD_0.05_ indicates least significant difference values calculated at the α = 0.05 level. a, b, c, d, and e were used to indicate the significant level of *p* = 0.05.

## Data Availability

Not applicable.

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
