# Peer review of "Effects of Different Gene Editing Modes of CRISPR/Cas9 on Soybean Fatty Acid Anabolic Metabolism Based on GmFAD2 Family"

_ijms, 2023, doi:10.3390/ijms24054769_

Round 1

Reviewer 1 Report

-Detailed informations should be given about soybean material

-GC conditions must be specified exactly in determining the fatty acid content in soybean seeds.

-There are many typos in the article, they should be corrected.

-Conlusion section should be added..

Reviewer 2 Report

The review of the paper submitted to IJMS by Zhou et al. entitled: “Effects of different gene editing modes of CRISPR/Cas9 on soybean fatty acid anabolic metabolism based on GmFAD2 family”.

The main idea of the paper is to explore the breeding effects of gene editing produced by different mutation types to provide new ideas for improving the efficiency of plant gene editing and developing base editing and precision editing in the future. However the authors for such stated hypothesis of this research add a citation of their previous work, which lead me to the question about the real aim of this work. Have they already answer the question stated at the end of introduction in 2020?

The most serious remarks concern figures and documentation of the experiments. In the main file there is no figures. All figures are in the files original Images for Blots/Gels as well as in supplementary and non-published materials. In such situation I doubts what Figures would be presented by the authors. Moreover in supplementary files the figures are not describe so I am not sure what is presented on them.

As the Figures are not supplemented to the text and are not described I can not verified results presented in the paper. There is lack of results for determination of the fatty acid content in soybean seeds and analysis of agronomic traits, which in my opinion are stated as a clue of this paper.

There is also lack of description of some methods used in the study. How the authors obtained the transformed plants. There is already description how the authors obtain the transformed E. coli culture, but not how they obtained T0 plants. Also the point 4.5 is described very shortly without details of the methods applied.

The language needs to be verified as some of the sentences should be rewrite for proper meaning. F.eg. in L32 “soybean contains 17% to 22% of seed oil” should be like soybean seeds contain 17-22% of oil. L90 “positive plants were analyzed according to the gene editing breeding analysis process” I suppose it means the positively verified plants were or something else. I do not know what is show on Figure 2.

The text suffer from many typos and editorial mistakes. In my opinion it is impossible to prepare the reasonable review and verified and judge about results quality. That is why I advised to reject and advise to submitted once more after substantial correction.

Reviewer 3 Report

Manuscript "Effects of different gene editing modes of CRISPR/Cas9 on soybean fatty acid anabolic metabolism based on GmFAD2 family" is very interesting.

General comments:
Authors explored the breeding effects of gene editing produced by different mutation types to provide new ideas for improving the efficiency of plant gene editing and developing base editing and precision editing in the future.

The manuscript lacks statistical analysis of the results obtained, which disqualifies it as a scientific paper.

Paper needs major revision.

Round 2

Reviewer 2 Report

The authors modified the papers, supplemented all figures and described them. They modified just this sentences, which I indicated in the review. Thus in my opinion the paper need some minor revision. In my opinion there are still some typos. Please check whole the text carefully.

Why in Figure 3c there is no band for “+” line and why on Figure 5 the “+” is empty line?

On Figure 7 the authors have presented partial prediction of protein structure, however in M&M section there is no description of the methods applied for structure prediction. The authors should at least mention the program/software and methodology used in this prediction.

Author Response

Please see attached for details of changes.

Reviewer 3 Report

Table 2: Letter designation of homogeneous groups should be after average values.

Table 2 contains average values and some more results. Which ones? Sd or se? Clarification should be made.

Table 2 needs to be supplemented with LSD or HSD values.

Table 3: Letter designation of homogeneous groups should be after mean values.

Table 3 contains mean values and some more results. Which ones? Sd or se? Should be clarified.

Table 3 needs to be supplemented with LSD or HSD values.

No information on the correlation of the observed characteristics.
The manuscript needs to be supplemented with a subsection describing the statistical methods used.

Paper needs major revision.

Author Response

(The authors gave the same response as above.)

Round 3

Reviewer 2 Report

I trust the answers and appropriate changes made by authors.

Reviewer 3 Report

Table 2: Letter designation of homogeneous groups should be after average values.

Table 3: Letter designation of homogeneous groups should be after mean values.

Table 4: Letter designation of homogeneous groups should be after mean values.

Table 5: Letter designation of homogeneous groups should be after mean values.

The manuscript still lacks a "Statistical Analysis" subsection with a description of the statistical methods used.

Round 4

Reviewer 3 Report

Now, all is ok.